# PC2WF: 3D Wireframe Reconstruction from Raw Point Clouds

**Yujia Liu, Stefano D'Aronco, Konrad Schindler, Jan Dirk Wegner**
EcoVision Lab, Photogrammetry and Remote Sensing, ETH Zürich
`{firstname.lastname}@geod.baug.ethz.ch`

## Abstract

We introduce PC2WF, the first end-to-end trainable deep network architecture to convert a 3D point cloud into a wireframe model. The network takes as input an unordered set of 3D points sampled from the surface of some object, and outputs a wireframe of that object, i.e., a sparse set of corner points linked by line segments. Recovering the wireframe is a challenging task, where the numbers of both vertices and edges are different for every instance, and a-priori unknown. Our architecture gradually builds up the model: It starts by encoding the points into feature vectors. Based on those features, it identifies a pool of candidate vertices, then prunes those candidates to a final set of corner vertices and refines their locations. Next, the corners are linked with an exhaustive set of candidate edges, which is again pruned to obtain the final wireframe. All steps are trainable, and errors can be backpropagated through the entire sequence. We validate the proposed model on a publicly available synthetic dataset, for which the ground truth wireframes are accessible, as well as on a new real-world dataset. Our model produces wireframe abstractions of good quality and outperforms several baselines.

## 1 Introduction

Many practical 3D sensing systems, like stereo cameras or laser scanners, produce unstructured 3D point clouds. That choice of output format is really just a "smallest common denominator", the least committal representation that can be reliably generated with low-level signal processing. Most users would prefer a more efficient and more intuitive representation that describes the scanned object's geometry as a compact collection of geometric primitives, together with their topological relations. Specifically, many man-made objects are (approximately) polyhedral and can be described by corners, straight edges and/or planar surfaces. Roughly speaking there are two ways to abstract a point cloud into a polyhedral model: either find the planar surfaces and intersect them to find the edges and corners, e.g., Schnabel et al. (2007); Fang et al. (2018); Coudron et al. (2018); or directly find the salient corner and/or edges, e.g., Jung et al. (2016); Hackel et al. (2016).

A wireframe is a graph representation of an object's shape, where vertices correspond to corner points (with high Gaussian curvature) that are linked by edges (with high principal curvature). Wireframes are a good match for polyhedral structures like mechanical parts, furniture or building interiors. In particular, since wireframes focus on the edge structure, they are best suited for piece-wise smooth objects with few pronounced crease edges (whereas they are less suitable for smooth objects without defined edges or for very rough ones with edges everywhere). We emphasise that wireframes are not only a "compression technique" to save storage. Their biggest advantage in many applications is that they are easy to manipulate and edit, automatically or interactively in CAD software, because they make the salient contours and their connectivity explicit. Reconstructed wireframes can drive and help to create 3D CAD models for manufacturing parts, metrology, quality inspection, as well as visualisation, animation, and rendering.

Inferring the wireframe from a noisy point cloud is a challenging task. We can think of the process as a sequence of steps: find the corners, localise them accurately (as they are not contained in the point cloud), and link them with the appropriate edges. However, these steps are intricately correlated. For example, corner detection should "know" about the subsequent edge detection: curvature is affected by noise (as any user of an interest point detector can testify), so to qualify as a corner

a 3D location should also be the plausible meeting point of multiple, non-collinear edges. Here we introduce PC2WF, an end-to-end trainable architecture for point cloud to wireframe conversion. PC2WF is composed of a sequence of feed-forward blocks, see Fig. 1 for the individual steps of the pipeline. First a "backbone" block extracts a latent feature encoding for each point. The next block identifies local neighbourhoods ("patches") that contain corners and regresses their 3D locations. The final block links the corners with an overcomplete set of plausible candidate edges, collects edge features by pooling point features along each edge, and uses those to prune the candidate set to a final set of wireframe edges. All stages of the architecture that have trainable parameters support back-propagation, such that the model can be learned end-to-end. We validate PC2WF with models from the publicly available ABC dataset of CAD models, and with a new furniture dataset collected from the web. The experiments highlight the benefit of explicit, integrated wireframe prediction: on the one hand, the detected corners are more complete and more accurate than with a generic corner detector; on the other hand, the reconstructed edges are more complete and more accurate than those found with existing (point-wise) edge detection methods.

## 2 RELATED WORK

Relatively little work exists on the direct reconstruction of 3D wireframes. We review related work for the subtasks of corner and edge detection, as well as works that reconstruct wireframes in 2D or polyhedral surface models and geometric primitives in 3D.

**Corner Detection.** Corner points play a fundamental role in both 2D and 3D computer vision. A 3D corner is the intersection point of $\geq 3$ (locally) non-collinear surfaces. Corners have a well-defined location and a local neighbourhood with non-trivial shape (e.g., for matching). Collectively the set of corners often is sufficient to tightly constrain the scale and shape of an object. Typical local methods for corner detection are based on the $2^{nd}$-moment matrix of the points (resp. normals) in a local neighbourhood (Głomb, 2009; Sipiran & Bustos, 2010; 2011), a straight-forward generalisation of the 2D Harris operator first proposed for 2D images (Harris et al., 1988). Also other 2D corner detectors like Susan (Smith & Brady, 1997) have been generalised to 3D (Walter et al., 2009).

**Point-level Edge Point Detection.** A large body of work is dedicated to the detection of 3D points that lie on or near edges. The main edge feature is high local curvature (Yang & Zang, 2014). Related features, such as symmetry (Ahmed et al., 2018), change of normals (Demarsin et al., 2007), and eigenvalues (Bazazian et al., 2015; Xia & Wang, 2017) can also be used for edge detection. Hackel et al. (2016; 2017) use a combination of several features as input for a binary edge-point classifier. After linking the points into an adjacency graph, graph filtering can be used to preserve only edge points (Chen et al., 2017). EC-Net (Yu et al., 2018) is the first method we know of that detects edge points in 3D point clouds with deep learning.

**Learning Structured Models.** Many methods have been proposed to abstract point clouds into more structured 3D models, e.g., triangle meshes (Lin et al., 2004; Remondino, 2003; Lin et al., 2004), polygonal surfaces (Fang et al., 2018; Coudron et al., 2018), simple parametric surfaces (Schnabel et al., 2007; 2008; Li et al., 2019). From the user perspective, the ultimate goal would be to reverse-engineer a CAD model (Gonzalez-Aguilera et al., 2012; Li et al., 2017b; Durupt et al., 2008). Several works have investigated wireframe parsing in 2D images (Huang et al., 2018; Huang & Gao, 2019). LCNN (Zhou et al., 2019a) is an end-to-end trainable system that directly outputs a vectorised wireframe. The current state-of-the-art appears to be the improved LCNN of Xue et al. (2020), which uses a holistic "attraction field map" to characterise line segments. Zou et al. (2018) estimate the 3D layout of an indoor scene from a single perspective or panoramic image with an encoder-decoder architecture, while Zhou et al. (2019b) obtain a compact 3D wireframe representation from a single image by exploiting global structural regularities. Jung et al. (2016) reconstruct 3D wireframes of building interiors under a Manhattan-world assumption. Edges are detected heuristically by projecting the points onto 2D occupancy grids, then refined via least-squares fitting under parallelism/orthogonality constraints. Wireframe extraction in 2D images is simplified by the regular pixel grid. For irregular 3D points the task is more challenging and there has been less work.

Another related direction of research is 3D shape decomposition into sets of simple, parameterised primitives (Tulsiani et al., 2017; Li et al., 2017a; Zou et al., 2017; Sharma et al., 2018; Du et al., 2018). Most approaches use a coarse volumetric occupancy grid as input for the 3D data, which

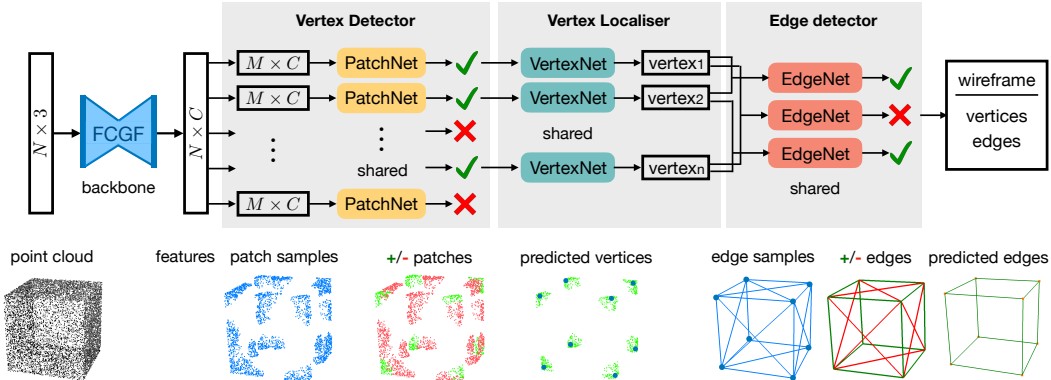

Figure 1: Illustration of our wireframe modelling architecture.

works well for shape primitive fitting but only allows for rather inaccurate localisation of edges and vertices. Li et al. (2019) propose a method for fitting 3D primitives directly to raw point clouds. A limiting factor of their approach is that only a fixed maximum number of primitive instances can be detected. This upper bound is hard-coded in the architecture and prevents the network from fitting more complex structures. Nan & Wonka (2017) propose Polyfit to extract polygonal surfaces from a point cloud. Polyfit first selects an exhaustive set of candidate surfaces from the raw point cloud using RANSAC, and then refines the surfaces. Similarly to our PC2WF, Polyfit works best with objects made of straight edges and planar surfaces. The concurrent work Wang et al. (2020) proposed a deep neural network that is trained to identify a collection of parametric edges.

In summary, most 3D methods are either hand-crafted for restricted, schematic settings, or only learn to label individual "edge points". Here we aim for an end-to-end learnable model that directly maps a 3D point cloud to a vectorised wireframe.

## 3 METHOD

Our full architecture is depicted in Fig. 1. In the following we describe each block and, where necessary, discuss training and inference procedures. We denote a point cloud of $N$ 3D points with $\mathcal{P} = \{p_i\}_{i=1}^N$ where $p_i \in \mathbb{R}^3$ represents the 3D coordinates of point $i$. The 3D wireframe model of the point cloud $\mathcal{P}$ is denoted by $\mathcal{W} = (\mathcal{V}, \mathcal{E})$ and it is defined, analogously to an undirected graph, by a set of vertices $\mathcal{V}$ which are connected by a set of edges $\mathcal{E} \subseteq \mathcal{V} \times \mathcal{V}$. As the vertices represent points in the 3D space we can think of them as being described by 3D coordinates $v_i \in \mathcal{V} \subset \mathbb{R}^3$.

**Overall Pipeline.** Our feed-forward pipeline consists of four blocks: (1) a convolutional *backbone* that extracts per-point features; (2) a *vertex detector* that examines patches of the point cloud and decides which ones contain vertices; (3) a *vertex localiser* that outputs the location of the vertex within a patch; (4) an *edge detector* that instantiates candidate edges and prunes them to a wireframe.

**Backbone.** PC2WF first extracts a feature vector per 3D point that encodes its geometric context. Our pipeline is compatible with any (hand-coded or learned) function that outputs a fixed-size descriptor vector per point. We use Fully Convolutional Geometric Features (FCGF, Choy et al., 2019b), which are compact and capture broad spatial context. The backbone operates on sparse tensors (Choy et al., 2019a) and efficiently computes 32-dimensional features in a single, 3D fully-convolutional pass.

**Vertex Detector.** This block performs binary classification of local *patches* (neighbourhoods in the point cloud) into those (few) that contain a corner and those that do not. Only patches detected to contain a corner are passed on to the subsequent block. While one could argue that detecting the presence of a vertex and inferring its 3D coordinates are two facets of the same problem, we found it advantageous to separate them. Detecting first whether a patch is likely to contain a corner, then constructing its 3D coordinates reduces the computational cost, as we discard patches without corners early. Empirically it also reduces false positives and increases localisation accuracy.

Our patch generation is similar to EC-Net (Yu et al., 2018). A patch contains a fixed number $M$ of points, chosen by picking a patch centroid and finding the closest $(M-1)$ points in terms of geodesic distance on the underlying surface. The geodesic distance avoids "shortcuts" in regions with thin structures or high curvature. To approximate the geodesic distance, the entire point cloud is linked into a $k$-nearest neighbour graph, with edge weights proportional to Euclidean distance. Point-to-point distances are then found as shortest paths on that graph. During training we randomly draw positive samples that contain a corner and negative ones that do not. To increase the diversity of positive samples, the same corner can be included in multiple patches with different centroids. For the negative samples, we ensure to draw not only patches on flat surfaces, but also several patches near edges, by doing this negative samples with strong curvature are well represented. For inference, patch seeds are drawn using farthest point sampling. The vertex detector takes as input the (indexed) features of the $M$ points in the patch, max-pooled per dimension to achieve permutation invariance, and outputs a scalar *vertex/no_vertex* probability. Its loss function $\mathcal{L}_{pat}$ is binary cross-entropy.

**Vertex Localiser.** The vertex detector passes on a set of $P$ patches predicted to contain a corner. The $M$ points $p_i$ of such a patch form the input for the localisation block. The backbone features $f_i \in \mathbb{R}^C$ and the 3D coordinates $p_i \in \mathbb{R}^3$ of all $M$ points are concatenated and fed into a multi-layer perceptron with fully connected layers, ReLU activations and batch normalisation. The MLP outputs a vector of $M$ scalar weights $w_i$, where $\sum_i w_i = 1$, such that the predicted vertex location is the weighted mean of the input points, $v = \sum w_i p_i$. The prediction $v_j$ is supervised by a regression loss $\mathcal{L}_{vert} = \frac{1}{P} \sum_i^P \|v_i - v_i^{GT}\|_2^2$, summed over all $P$ corner patches. Predicting weights $w_i$ rather than directly the coordinates $\tilde{x}$ safeguards against extrapolation errors on unusual patches. We note that the localiser not only improves the corner point accuracy but also improves efficiency, by making it possible to use farthest point sampling rather than testing every input point.

**Edge Detector.** The edge detection block serves to determine which pairs of (predicted) vertices are connected by an edge. Candidate edges $e_{ij}$ are found by linking two vertices $v_i$ and $v_j$, see below. To obtain a descriptor for an edge, we sample the $N_s$ equally spaced query locations $\mathcal{Q} = \{q_k\}_{k=0}^{s-1}$ along the line segment between the two vertices, including the endpoints: $q_0 = \tilde{v}_i, q_{s-1} = \tilde{v}_j$. For each $q_k$ we find the nearest 3D input point $p_{NN(k)}$ (by Euclidean distance) and retrieve its backbone encoding $f_{NN(k)}$. This yields an $(N_s \times C)$ edge descriptor, which is max-pooled along its first dimension, inspired by the ROI pooling in Faster-RCNN (Ren et al., 2015) and LOI pooling in LCNN (Zhou et al., 2019a). The resulting $(\frac{1}{\text{stride}} N_s \times C)$ descriptor is flattened and fed through an MLP with two fully connected layers, which returns a scalar *edge/no_edge* score.

During training we select a number of vertex pairs from both the ground truth vertex list $\{v_i\}_{i=1}^G$ and the predicted vertex list $\{\tilde{v}_i\}_{i=1}^K$, and draw positive and negative sets of edge samples, $\mathcal{E}^+, \mathcal{E}^-$. We empirically found that evaluating all vertex pairs contained in $\mathcal{E}^+, \mathcal{E}^-$ during training leads to a large imbalance between positive and negative samples. Similar to Zhou et al. (2019a), we thus draw input edges for the edge detector from the following sets:

**(a) Positive edges between ground truth vertices:** This set comprises all true edges in the ground truth wireframe $\mathcal{E}^{\text{gt+}} = \mathcal{E}$

**(b) Negative edges between ground truth vertices:** Two situations are relevant: "spurious edges", i.e., connections between two ground truth vertices that do *not* share an edge (but lie on the same surface, such that there are nearby points to collect features). And "inaccurate edges" where one of the endpoints is not a ground truth vertex, but not far from one (to cover difficult cases not far from a correct edge). Formally:

$$\mathcal{E}^{\text{gt-}} = \{e_{v_i, v_j} = (v_i, v_j) \mid v_i, v_j \in V, e_{ij} \notin \mathcal{E}, \forall p \text{ on } e_{v_i, v_j}, \exists p' \in \mathcal{P}, \|p - p'\|_2 < \epsilon\}$$
$$\cup \{e_{i,p_k} = (v_i, p_k) \mid (v_i, v_j) \in \mathcal{E}, p_k \in \mathcal{P}, \epsilon_1 \leq \|v_j - p_k\|_2 \leq \epsilon_2\}$$

**(c) Positive edges between predicted vertices:** Connections between two "correct" predicted vertices that have both been verified to coincide with a ground truth wireframe vertex up to a small threshold $\epsilon_+$. Formally:

$$\mathcal{E}^{\text{pred+}} = \{e_{\tilde{v}_i, \tilde{v}_j} = (\tilde{v}_i, \tilde{v}_j) \mid \min \|\tilde{v}_i - v_k\|_2 < \epsilon_+, \min \|\tilde{v}_j - v_l\|_2 < \epsilon_+, (v_k, v_l) \in \mathcal{E}^{gt+}\}$$

**(d) Negative edges between predicted vertices:** These are *(i)* "wrong links" as in $\mathcal{E}^{\text{gt-}}$, between "correctly" predicted vertices; and *(ii)* "hard negatives" where exactly one of the two vertices is close

to a ground truth wireframe vertex, to cover "near misses".

$$\mathcal{E}^{\text{pred-}} = \{e_{\tilde{v}_i, \tilde{v}_j} = (\tilde{v}_i, \tilde{v}_j) \mid \min \|\tilde{v}_i - v_k\|_2 < \epsilon_-, \min \|\tilde{v}_j - v_l\|_2 < \epsilon_-, (v_k, v_l) \in \mathcal{E}^{gt-}\}$$
$$\cup \{e_{\tilde{v}_i, \tilde{v}_j} = (\tilde{v}_i, \tilde{v}_j) \mid \epsilon_1 \leq \|\tilde{v}_i - v_k\|_2 \leq \epsilon_2, \|\tilde{v}_j - v_l\|_2 \leq \epsilon_2, v_k, v_l \in \mathcal{V}\}$$

Fig. 2 visually depicts the different edge sets. To train the edge detector we randomly sample positive examples from $\mathcal{E}^+ = \mathcal{E}^{gt+} \cup \mathcal{E}^{pred+}$ and negative examples from $\mathcal{E}^- = \mathcal{E}^{gt-} \cup \mathcal{E}^{pred-}$. As loss function $\mathcal{L}_{edge}$ we use the balanced binary cross entropy. During inference, we go through the fully connected graph of edge candidates connecting any two (predicted) vertices. Candidates with a too high average distance to the input points (i.e., not lying on any object surface) are discarded. All others are fed into the edge detector for verification.

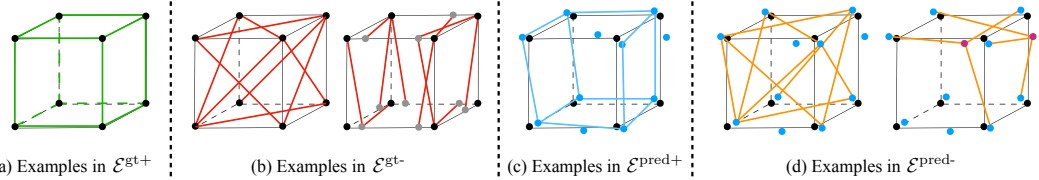

(a) Examples in $\mathcal{E}^{gt+}$    (b) Examples in $\mathcal{E}^{gt-}$    (c) Examples in $\mathcal{E}^{pred+}$    (d) Examples in $\mathcal{E}^{pred-}$

Figure 2: Positive and negative samples for edge verification during training. (a) Positive edge samples (green lines) between ground truth vertices (black dots). (b) "spurious" (left) and "inaccurate" (right) connections between ground truth vertices. (c) positive samples (blue lines) between predicted vertices (blue dots). (d) "wrong links" (left) and "near misses" (right) between predicted vertices.

**End-to-end training.** When training the complete network, we minimise the joint loss function with balancing weights $\alpha$ and $\beta$: $\mathcal{L} = \mathcal{L}_{pat} + \alpha \mathcal{L}_{vert} + \beta \mathcal{L}_{edge}$.

**Non-Maximum Suppression** (NMS) is a standard post-processing step for detection tasks, to suppress redundant, overlapping predictions. We use it *(i)* to prevent duplicate predictions of vertices that are very close to each other, and *(ii)* to prune redundant edges.

## 4    EXPERIMENTS

We could not find any publicly available dataset of point clouds with polyhedral wireframe annotations. Thus, we collect our own synthetic datasets: a *subset from ABC* (Koch et al., 2019) and a *Furniture dataset*. ABC is a collection of one million high-quality CAD models of mechanical parts with explicitly parameterised curves, surfaces and various geometric features, from which ground truth wireframes can be generated. We select a subset of models with straight edges. It consists of 3,000 samples, which we randomly split into 2,000 for training, 500 for validation, and 500 for testing. For the furniture dataset we have collected 250 models from Google 3DWarehouse for the categories *bed, table, chair/sofa, stairs, monitor* and render ground truth wireframes. Examples are shown in the supplementary material. To generate point clouds, we use the virtual scanner of Wang et al. (2017). We place 14 virtual cameras on the face centres of the truncated bounding cubes of each object, uniformly shoot 16,000 parallel rays towards the object from each direction, and compute their intersections with the surfaces. Finally, we jitter all points by adding Gaussian noise $\mathcal{N}(0, 0.01)$, truncated to $[-0.02, 0.02]$. The number $N$ of points varies between $\approx 100k - 200k$ for different virtually scanned point clouds. For details, see the supplementary material.

### 4.1    EVALUATION MEASURES

We first separately evaluate the vertex detection and edge detection performance of our architecture, then assess the reconstructed wireframes in terms of structural average precision and of a newly designed wireframe edit distance.

**Mean Average precision mAP$^v$ for vertices.** To generate high quality wireframes, the predicted vertex positions must be accurate. This metric evaluates the geometric quality of vertices, regardless of edge links. A predicted vertex is considered a true positive if the distance to the nearest ground truth vertex is below a threshold $\eta_v$. For a given $\eta_v$ we generate precision-recall curves by varying

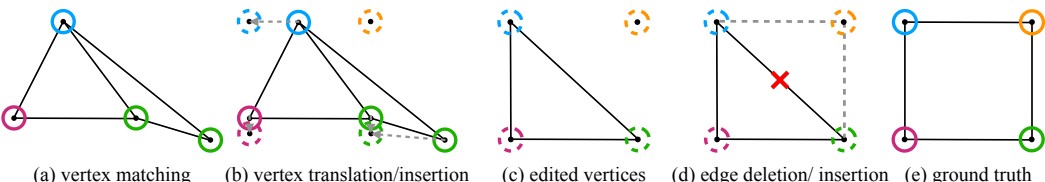

(a) vertex matching  (b) vertex translation/insertion  (c) edited vertices  (d) edge deletion/ insertion  (e) ground truth

Figure 3: Illustration of WED. (a) First match vertices to (e) ground truth; (b) move matched vertices, insert missing ones and compute the cost; (d) delete and insert edges and compute the cost.

the detection threshold. Average precision ($AP^v$) is defined as the area under this curve, $mAP^v$ is averaged over different thresholds $\eta_v$.

**Point-wise precision and recall for edges.** We do not have a baseline that directly extracts 3D wireframe edges. Rather, existing methods label individual input points as "lying on an edge", and produce corresponding precision-recall curves (Hackel et al., 2016; Yu et al., 2018). To use this metric, we label a 3D point as "edge point" if its distance to the nearest wireframe edge is below the average point-to-point distance. As for $AP^v$, we can now check whether a predicted "edge point" is correct up to $\eta_e$, and compute $AP^e$ and mean average precision ($mAP^e$) over different $\eta_e$.

**Structural average precision for wireframes.** The above two evaluation metrics looks at vertices and edges separately, in terms of geometry. In order to comprehensively evaluate the generated wireframe, we also need to take into account the connectivity. The structural average precision (sAP, Zhou et al., 2019a) metric, inspired by mean average precision, can be used to assess the overall prediction. This metric checks if the two end points $(\tilde{v}_i, \tilde{v}_j)$ of a detected edge both match the end points $(v_k, v_l)$ of a ground truth edge up to a threshold $\eta_w$, i.e., $\min_{v_k \in \mathcal{V}} \|\tilde{v}_i - v_k\|_2 + \min_{v_l \in \mathcal{V}} \|\tilde{v}_j - v_l\|_2 < \eta_w, (v_k, v_l) \in \mathcal{E}$. We report sAP for different $\eta_w$ and mean sAP (msAP).

**Wireframe edit distance (WED).** In order to topologically compare wireframes, we have designed a quality measure based on edit distance. We consider wireframes as geometric graphs drawn in 3D space. Distance measures commonly used in computational geometry and graph theory, such as Hausdorff distance and Frechet distance, resistance distance, or geodesic distance, do not seem to apply to our situation. Inspired by Cheong et al. (2009), we propose a wireframe edit distance (WED). It is a variant of the graph edit distance (GED), which measures how many elementary graph edit operators (vertex insertion/deletion, edge insertion/deletion, etc.) are needed to transform one graph to another. Instead of allowing arbitrary sequences of operations, which makes computing the GED NP-hard, we fix their order, see Fig. 3: (1) vertex matching; (2) vertex translation; (3) vertex insertion; (4) edge deletion; (5) edge insertion. Let the ground truth wireframe $\mathcal{W}^{gt} = (\mathcal{V}^{gt}, \mathcal{E}^{gt})$ and the predicted wireframe $\mathcal{W}^o = (\mathcal{V}^o, \mathcal{E}^o)$. First, we match each vertex $v_i^o$ in $\mathcal{W}^o$ to the nearest ground truth vertex $v_i^{gt}$ in $\mathcal{W}^{gt}$, then translate them at a cost of $c_v \cdot \|v_i^o v_u^{gt}\|$. New vertices are inserted where a ground truth vertex has no match (at no cost, as cost will accrue during edge insertion). After all vertex operations, we perform edge deletion and insertion based on ground truth edges at a cost of $c_e \cdot \|e\|$, where $\|e\|$ is the length of deleted/inserted edges. The final WED is the sum of all cost terms.

## 4.2 RESULTS AND COMPARISONS

**Vertex detection.** We validate the PC2WF vertex detection module and compare to a Harris3D baseline (with subsequent NMS to guarantee a fair comparison) on the ABC dataset, Tab. 1. Exemplary visual results and precision-recall curves are shown in Fig. 4 and Fig.5. Harris3D often fails to accurately localise a corner point, and on certain thin structures it returns many false positives. The patch size $M$ used in our vertex detector has an impact on performance. Patch sizes between 20 and 50 achieve the best performance, where the precision stays at a high level over a large recall range. Patch size 1 leads to low precision (although still better than Harris3D), confirming that per-point features, even when computed from a larger receptive field, are affected by noise and insufficient to find corners with acceptable false positive ratio. We refer the interested reader to the supplementary material for a more exhaustive ablation study.

Table 1: Vertex detection results.

| Dataset | ABC | | | | furniture | | | |
|---|---|---|---|---|---|---|---|---|
| Method | $AP^v_{0.02}$ | $AP^v_{0.03}$ | $AP^v_{0.05}$ | mAP | $AP^v_{0.02}$ | $AP^v_{0.03}$ | $AP^v_{0.05}$ | mAP |
| Harris3D + NMS | 0.262 | 0.469 | 0.733 | 0.488 | 0.268 | 0.460 | 0.712 | 0.480 |
| **our method** | **0.856** | **0.901** | **0.925** | **0.894** | **0.847** | **0.911** | **0.924** | **0.894** |

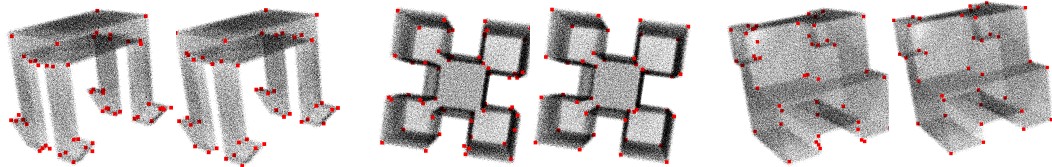

Figure 4: Vertex detection results of Harris3D (left) and ours (right) on the ABC dataset.

**Edge prediction.** Due to the limited number of methods able to extract wireframes from 3D point clouds, we compare PC2WF to recent algorithms that detect edge points, such as FREE (Bazazian et al., 2015), ECD (Ahmed et al., 2018), FEEM (Xia & Wang, 2017), Contour Detector (Hackel et al., 2016) and EC-Net (Yu et al., 2018). We also compare with Polyfit (Nan & Wonka, 2017), which is the only method able to reconstruct a vectorial representation similar to ours from a point cloud. Since Polyfit extracts planar surfaces of the object, we must convert that representation into a wireframe, which amounts to identifying the plane intersections. Precision-recall curves for all methods are shown in Fig. 6. Visual results are shown in Fig. 8. PC2WF outperforms all baselines based on hand-crafted features like FREE, ECD, FEEM and Contour Detector, as well as the deep learning method EC-Net, across all quality measures (Tab. 2). A visual comparison between the ground truth, PC2WF, EC-Net Yu et al. (2018), and Polyfit is shown in Fig. 9. One can clearly see the advantage of predicting vector edges rather than "edge points", as they lead to a much sharper result. The inevitable blur of edge points across the actual contour line, and their irregular density, make vectorisation a non-trivial problem. Polyfit and PC2WF show comparable performance for this specific point cloud. PC2WF failed to localise some corners and missed some small edges, whereas Polyfit failed to reconstruct the lower part of the armchair.

**Wireframe Reconstruction.** We visualise PC2WF's output wireframes in Fig. 10. Structural precision-recall curves with various patch sizes are shown in Fig. 7. For additional examples of output predictions, including cases with curved edges, please refer to the supplementary material. The results are aligned with the ones of the vertex module. Moderate patch sizes between 20 and 50 achieve the best performance. Tab. 3 shows the structural average precision with patch size 50, which exceeds the one of Polyfit. The results in terms of the wireframe edit distance are shown in Tab. 4 for PC2WF and Polyfit. The average number of predicted vertices (21.7) and ground truth vertices (20.1) per object are close, such that only few vertex insertions are needed in most cases. Although all predicted vertices need some editing, the $WED_v$ caused by those edits is small (0.2427), meaning that the predictions are accurate. On the contrary, the 2.6 edges that need to be edited on

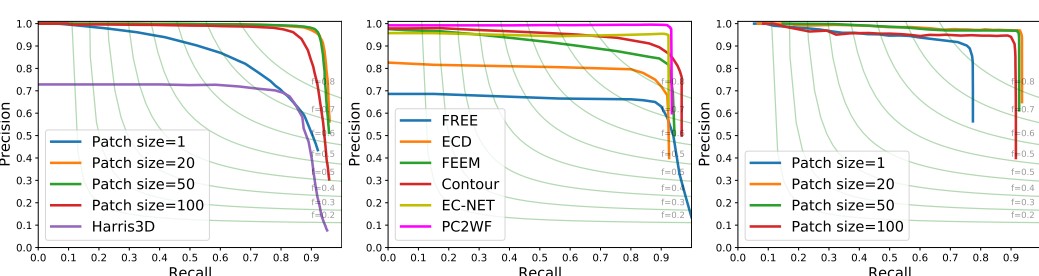

Figure 5: Vertex prediction accuracy on ABC dataset.

Figure 6: Edge detection accuracy on ABC dataset.

Figure 7: Wireframe accuracy on ABC dataset.

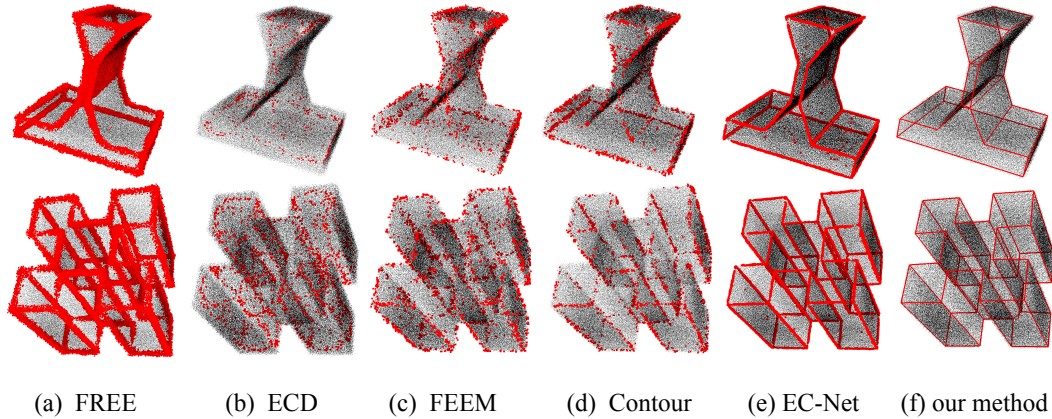

| (a) FREE | (b) ECD | (c) FEEM | (d) Contour | (e) EC-Net | (f) our method |

Figure 8: Comparison with edge point detection algorithms.

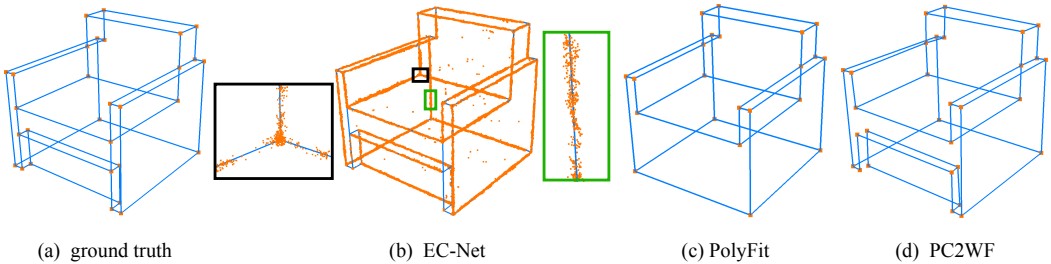

| (a) ground truth | (b) EC-Net | (c) PolyFit | (d) PC2WF |

Figure 9: Visual comparison of graph structures. Vertices are orange and edges are blue.

Table 2: Edge prediction results on the ABC subset and furniture dataset.

| Dataset | ABC | | | | furniture | | | |
|---|---|---|---|---|---|---|---|---|
| Method | $AP^e_{0.01}$ | $AP^e_{0.02}$ | $AP^e_{0.03}$ | $mAP^e$ | $AP^e_{0.01}$ | $AP^e_{0.02}$ | $AP^e_{0.03}$ | $mAP^e$ |
| FREE | 0.089 | 0.365 | 0.622 | 0.359 | 0.076 | 0.355 | 0.610 | 0.347 |
| ECD | 0.401 | 0.703 | 0.834 | 0.646 | 0.405 | 0.699 | 0.840 | 0.648 |
| FEEM | 0.412 | 0.754 | 0.870 | 0.679 | 0.419 | 0.760 | 0.865 | 0.681 |
| Contour | 0.422 | 0.812 | 0.913 | 0.715 | 0.420 | 0.809 | 0.900 | 0.710 |
| PolyFit | 0.771 | 0.798 | 0.882 | 0.817 | 0.584 | 0.667 | 0.728 | 0.660 |
| EC-Net | 0.503 | 0.839 | 0.912 | 0.751 | 0.499 | 0.838 | 0.924 | 0.750 |
| **our method** | **0.826** | **0.907** | **0.931** | **0.888** | **0.851** | **0.951** | **0.974** | **0.925** |

Table 3: sAP evaluation on two datasets.

| Dataset | ABC | | | | furniture | | | |
|---|---|---|---|---|---|---|---|---|
| Method | $sAP^{0.03}$ | $sAP^{0.05}$ | $sAP^{0.07}$ | msAP | $sAP^{0.03}$ | $sAP^{0.05}$ | $sAP^{0.07}$ | msAP |
| Polyfit | 0.753 | 0.772 | 0.781 | 0.769 | 0.552 | 0.590 | 0.612 | 0.585 |
| **ours** | **0.868** | **0.898** | **0.907** | **0.891** | **0.865** | **0.901** | **0.925** | **0.897** |

average increase $WED_e$ a lot more (1.4142) and dominate the WED, i.e., the inserted/deleted edges have significant length.

## 5 CONCLUSION

We have proposed PC2WF, an end-to-end trainable deep architecture to extract a vectorised wireframe from a raw 3D point cloud. The method achieves very promising results for man-made, polyhedral

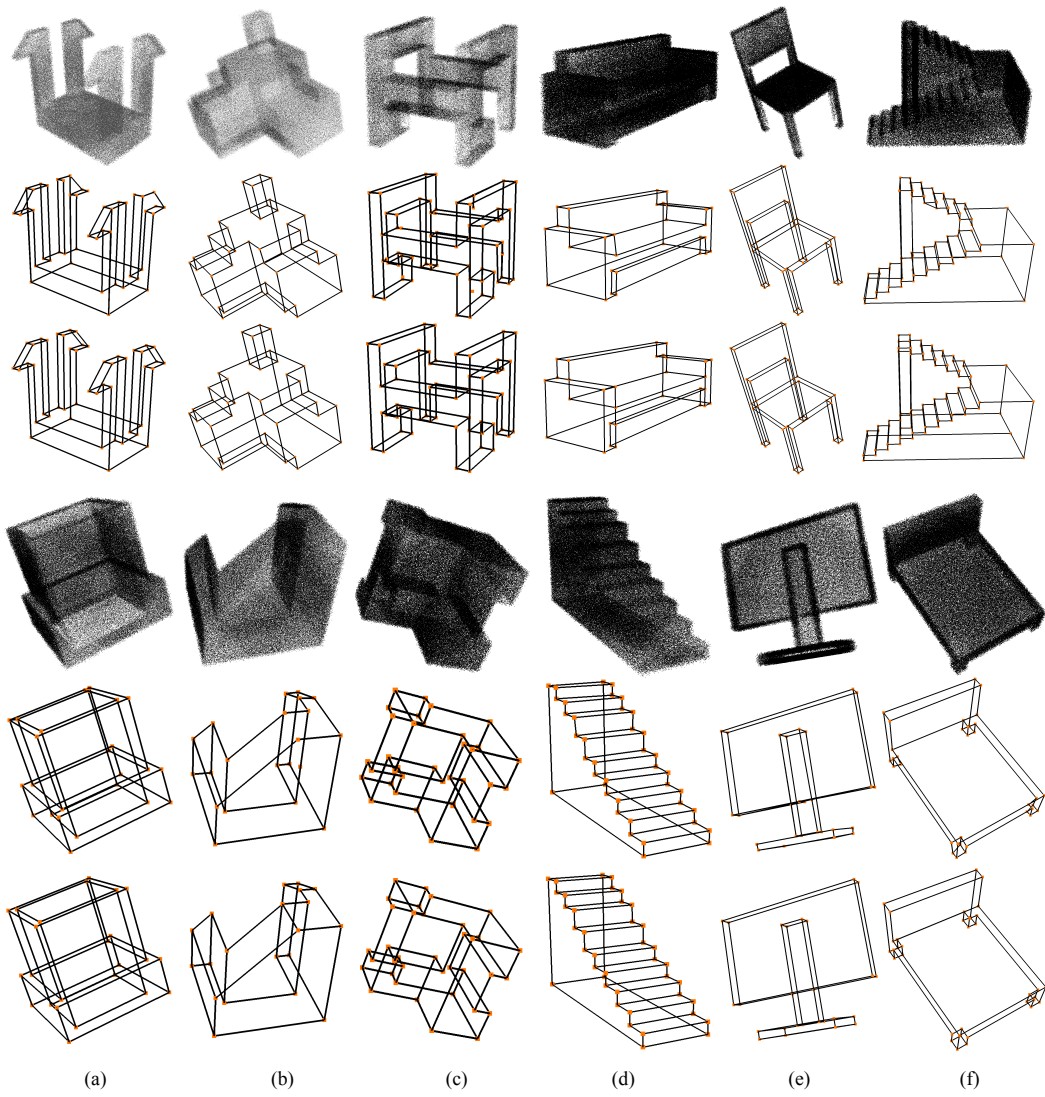

Figure 10: Wireframe reconstruction results on ABC (a)(b)(c) and furniture (d)(e)(f) datasets. Top: input raw point clouds. Middle: predicted wireframe. Bottom: ground truth.

Table 4: Wireframe edit distance (WED) on ABC dataset.

|          | pred #v | gt #v | edited #v | WED$_v$ | pred #e | gt #e | edited #e | WED$_e$ | WED |
|----------|---------|-------|-----------|---------|---------|-------|-----------|---------|--------|
| Polyfit  | 14.6    | 20.1  | 14.6      | **0.1251** | 21.9  | 30.2  | 10.2      | 4.7202  | 4.8453 |
| **ours** | 21.7    | 20.1  | 21.7      | 0.2427  | 32.8    | 30.2  | 2.6       | **1.4142** | **1.6569** |

objects, going one step further than low-level corner or edge detectors, while at the same time outperforming them on the isolated vertex and edge detection tasks. We see our method as one further step from robust, but redundant low-level vision to compact, editable 3D geometry. As future work we aim to address the biggest limitation of our current method which is the inability to handle wireframes with strongly curved edges.

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
