# OpenReview forum: "PC2WF: 3D Wireframe Reconstruction from Raw Point Clouds"
_ICLR.cc/2021/Conference — ICLR 2021 Poster_

### Official Review · AnonReviewer2 · 2020-10-19
**This paper solves a novel problem, provides a comprehensive benchmark, and shows good results. I recommend acceptance, providing that the authors address all my comments.**

**Rating:** 7
**Confidence:** 4

**Review:**

This paper introduces a supervised neural network predicting a wireframe structure from a 3D point cloud. The network takes a raw unordered 3D point cloud as input, processes it using FCGF architecture, and predicts three types of information: vertex existence in each patch, vertex location, and edge existence for each pair of vertices. In the experiments, the network is evaluated with two datasets, a subset of the ABC dataset and a set of 3D models from Google 3D warehouse. Also, it is compared with the baseline methods using four evaluation metrics, which are created to assess the accuracy of the predicted vertices, edges, and the overall wireframe graph structure. The results demonstrate the outperformance of the proposed method quantitatively and qualitatively.

*** Strengths ***
To my knowledge, this is the first work of leveraging neural networks to predict the wireframe structure from a 3D point cloud, except for the following concurrent work:
Wang et al., PIE-NET: Parametric Inference of Point Cloud Edges, arXiv:2007.04883.
While the architecture is simple, the proposed network shows the outperformance compared with the other non-learning-based methods as well as a previous edge detection network called EC-Net (Yu et al., 2018). The experiments are also carefully designed with two datasets and four evaluation metrics; the metrics include the precision of vertex prediction, edge prediction, and the entire graph structure prediction. The supplementary material also shows the details of network training and the performance changes depending on the choice of positive/negative edge sampling methods in the training.

*** Weaknesses ***
As mentioned in the conclusion of the paper, the biggest limitation is that the proposed method can only predict straight lines as edges. Also, the experiments are conducted only on the synthetic data but not on any real scan data. Some expositions are also not clear to me, and I describe the details of my concerns below in 'Questions and Suggestions'.

*** Questions and Suggestions ***
- In the patch generation and vertex location prediction, how is it guaranteed that each patch has only one vertex? The vertex localizer cannot predict multiple vertices since it predicts the location as a weighted sum.
- In the vertex localizer, wouldn't it be possible to combine binary classification loss for each point?
- In the end-to-end training, if the network makes false positives of vertex detection (especially at the beginning of the training) and thus the predicted patch does not have a GT vertex, how is the vertex localizer trained?
- In Section A.3 in supplementary, how is the backbone FCGF pretrained?
- While the authors described that the network is trained end-to-end, wouldn't it be better to pretrain each of the three components separately and finetune all of them in the end-to-end fashion? Since the later components take the outputs of the previous components in the end-to-end training, I guess the later components can be biased by some wrong predictions at the beginning of the training.
- For the vertex prediction, is the consistency across the patches checked? If a vertex is predicted in a region where two or more patches are overlapped, one can check whether all the overlapped patches predict the same vertex.
- For the edge prediction, wouldn't it be also possible to check whether the predicted edge lies on the input point cloud? One can sample points over the predicted edge and compute the distance from the sampled points to the closest points in the input point cloud.

*** Justification ***
I think this work is worth to be published in ICLR in that it first proposed the problem of learning for wireframe prediction in 3D point clouds. This is a fundamental problem in computer vision/graphics not only for understanding the structure of the raw geometry but also for practically converting raw 3D scanned data into an editable form. While the proposed network is simple, the supervised approach outperformed all the classic non-learning methods in the experiments. The benchmarks including datasets and evaluation metrics will also inspire future research in this direction.


I hope the authors discuss the following concurrent work in the revision:
Wang et al., PIE-NET: Parametric Inference of Point Cloud Edges, arXiv:2007.04883.


In short, I'll vote for acceptance once the authors address all my concerns above.

*** Minor Comments ***
- I could not find which numbers are used for parameters in the experiments: e.g., the number of points N, the number of patches M, and the weights in the loss function alpha and beta.
- I suggest changing the title of Section C in the supplementary to 'Choice of Positive and Negative Edge Sampling' or a similar one.
- In all tables, it would be good if 1) the best results are highlighted in bold (as done in Table 2) and 2) the caption describes which evaluation metric results are shown in the table.

---

> ### Author Response · Authors · 2020-11-19
> **Response to Reviewer2**
>
> We thank the reviewer for the valuable suggestions. We answer all comments in detail below and add additional text, experiments, and illustrations to the main paper as well as to the supplementary material. All changes are highlighted in blue.
>
> * During training, we generate patches with a vertex as centroid, but reject and replace patches that would contain another ground truth vertex, to avoid passing misleading evidence to the network. During inference, the vertex localiser only predicts one vertex per patch, even if this patch contains multiple vertices. This strategy indeed does not guarantee that all corners are detected, but since patches may partially overlap, the vertex localiser can still detect multiple points close to eachother. Empirically, we did not observe significant problems due to this strategy.
>
> * Using a binary classification for the vertex localiser would mean that one of the original points must be chosen. However, the exact corner are normally not contained in the point cloud. Therefore, we prefer a regression loss that can generate new, more accurate vertex coordinates.
>
> * We only use true positive predictions of the vertex detector for training the subsequent vertex localiser.
>
> * The FCGF architecture captures a lot of spatial context, and is trained with metric learning losses for fully-convolutional feature learning. We use the authors' pre-trained model to extract point-wise features and fine-tune it during our end-to-end training.
>
> * It is certainly feasible to perform a first, separate training. We initially did this experiment and obtained good performance, but sought a more elegant, end-to-end trainable solution. In order to avoid negative impacts on some network modules during training, we make sure that bad predictions of the previous module are not forwarded to later stages. For instance, the vertex localiser uses only true positive predictions during training. E.g., if vertex prediction is inaccurate (which is likely at early training stages), edge detection can still proceed successfully when fed only edges between ground truth vertices.
>
> * We did not check the consistency across patches for vertex prediction. Vertex prediction is supervised by the regression loss, and thus only very few vertices are predicted multiple times. Predicted vertices are likely to cluster and we thus use non-maximum suppression to prevent duplicate prediction of very close vertices from overlapping patches. In addition, duplicate or false positive vertices do not make it to the final wireframe, because the edge detector will parse the vertex candidates to create the edges. So the subsequent step that uses the vertices checks them and serves as a second layer of safety against wrong vertex predictions.
>
> * For the edge detection, we first check if an edge candidate lies on the input point cloud with reasonable accuracy, instead of simply verifying every edge in a fully connected graph.
>
> * The concurrent PIE-NET is a deep neural network that is trained to identify parametric edges. Although that paper and ours serve similar purposes, we look at the task from different perspectives and use different methods to solve it. PIE-NET extracts the corner and edge points in the first stage and then infers a collection of parametric edges based on them. Our wireframe is undirected, unweighted graph. We first detect a set only vertices which represent corners, then make link predictions between vertices. We note that PIE-NET appeared in NeurIPS 2020, after our submission to ICLR. Nonetheless, we attempted a comparison. Unfortunately, we could not get the open-sourced PIE-NET code to work, despite spending several days attempting to fix it. We also found that training data is not provided for end-to-end training, but only for the edge/corner detection. We even considered generating new training data from scratch, but the PIE-NET repository provides neither code for data pre-processing nor data formatting specifications. Our only option is an indicative comparison to the ABC results in the PIE-NET. According to Fig.11 of that paper the highest precision and recall for edges, with noise-free training and test data, are 0.692 (prec), respectively 0.858 (rec). We achieve precision 0.952 and recall 0.913 for edges, see our Fig. 7 (patch_size=20). We emphasise that the comparison is not rigorous and we do not claim that the numbers are directly comparable.
>
> * In terms of parameters used, the number of points N varies between $\approx$100k--200k, depending on the surface area of the object. The patch size M in our experiments was 1, 20, 50, its impact on performance was discussed in Sec. 4.2 of the main paper and in the supplementary material. The weights in the loss function are alpha=10, and beta=1. We point those values out more clearly in the revised manuscript.
>
> * Thanks for the minor comments. We made all changes in the revision.

---

### Official Review · AnonReviewer1 · 2020-10-26
**first sound deep wireframe extraction from 3D PC**

**Rating:** 7
**Confidence:** 3

**Review:**

This paper presents a deep architecture to extract a wireframe model from a 3D point cloud. This is a problem of high interest, and the author claim that the approach they present is the first one to address this task, which is true to the best of my knowledge. Since both the approach and the evaluation are sound, this alone seem to warrant publication. There are however several weaknesses in the paper, and my accept recommendation is conditional to clear answer on each of them:

1. An important point is the dataset that is introduced: will it be made publicly available? If not, this is a real issue, since it would create a high-cost entrance barrier for any following paper working on the same problem + it would makes comparison between the work and any other method impossible. Similarly, making the code available to allow comparison would be good.

2. The paper and supplementary material show little qualitative results. I think results on a large nimber of random shapes from the test sets should be provided, it is otherwise very hard to figure out how well the pipeline is actually working.

3.  related work on learning 3D shape decomposition into primitives is a bit quick/lacking, I would like discussion and ideally comparison with: https://openaccess.thecvf.com/content_cvpr_2017/papers/Tulsiani_Learning_Shape_Abstractions_CVPR_2017_paper.pdf seem very relevant to me, also GRASS https://dl.acm.org/doi/pdf/10.1145/3072959.3073637 , https://openaccess.thecvf.com/content_ICCV_2017/papers/Zou_3D-PRNN_Generating_Shape_ICCV_2017_paper.pdf ,https://openaccess.thecvf.com/content_cvpr_2018/papers/Sharma_CSGNet_Neural_Shape_CVPR_2018_paper.pdf , https://dl.acm.org/doi/pdf/10.1145/3272127.3275006 . While I realize it is not direct, I find it hard to believe that comparison with none of these works was possible.

4. All the examples shown are with very dense point clouds and very little noise, an analysis of the influence of these two parameters would be necessary. Also, it's unclear how robust the method is to variations in sampling (with a common M parameter)

Some smaller concerns/comments:
- p4, the paper refers to "testing all vertex pairs" when describing the training, which is confusing. Also, the argument that using all pairs during training (as done during inference) would create an imbalance is not so clear since balanced BCE is used in the loss.
- the task seem very well adapted to unsupervised training, proposing a completely supervised approach and not testing any unsupervised one is a bit underwhelming
- to me, Li et al. 2019 would be the natural baseline, but it's not discussed in details/not compared to. Would a comparison be possible?

---

> ### Author Response · Authors · 2020-11-19
> **Response to Reviewer1 (Part 1/2)**
>
> We thank the reviewer for the valuable suggestions. We answer all comments in detail below and add additional text, experiments, and illustrations to the main paper as well as to the supplementary material. All changes are highlighted in blue.
>
> * We plan to release all code and the dataset after acceptance of the paper.
>
> * We included more qualitative results in the new version of the main manuscript as well as in the supplementary materials.
>
> * We agree that shape decomposition into primitives is an important field of structuring 3D objects and we are happy to add a discussion about the suggested papers in the related works section. However, that field of research significantly differs from the topic of wireframe extraction. For example, most of the mentioned papers do not use an unstructured point cloud as input, and it is thus very hard to compare to them in a meaningful way. Although many works use a voxel representation that could be generated from the raw point cloud, the 3D grid resolution is necessarily coarser and would fail to achieve good accuracy in terms of edge and vertex localisation. To the best of our knowledge, the only work that does allow for a somewhat meaningful comparison is SPFN of (Li et al.2019). We discuss and compare SPFN below and add text and results to the supplementary material.
>
> * The noise and the point cloud density obviously affect the performance of the wireframe prediction. If the density is reduced (or the noise is increased) to the extent that some of the geometric structure details are lost, those details will be missing from the wireframe, too. We are currently running further experiments in order to supply such evaluation. Provided that the experiments finish in time, we will add the results to the manuscript/supplementary material.
>
> * Regarding sampling during training: “we randomly draw positive M-point patch samples that contain a corner and negative ones that do not. To increase the diversity of positive samples, the same corner can be included in multiple patches with different centroids. For the negative samples, we ensure to draw not only patches on flat surfaces, but also several patches near edges.”
> This sampling method can guarantee that a balanced number of positive and negative samples, of different types, are drawn. If we would simply pick patch centroids randomly, we would most likely sample very few positive samples containing vertices. Moreover, we would get a large number of very similar negative patches lying on flat planes, and too few in other, more challenging regions. The method described in the paper is designed to sample a representative and heterogeneous set of patches that covers the relevant variability in the training (and test) set.
>
> * We have adapted the text in the revised paper to emphasise the difference between train and testing operations, sorry for the confusion. In fact, using all possible vertex pairs turned out to be a bad strategy: performance was poor even when using a balanced BCE loss, furthermore it was computationally very expensive.
>
> * Unsupervised training is indeed an exciting direction and will be part of our future work. However, in our view it made more sense to start with a supervised setting and then take the next step. We are not aware of any unsupervised methods in the literature that can be used for wireframe prediction from point clouds.

---

> ### Author Response · Authors · 2020-11-19
> **Response to Reviewer1 (Part 2/2)**
>
> * SPFN (Li et al.2019) is a neural network that can map a 3D point cloud to a number of geometric surface primitives that best fit the underlying shape. This is a fairly different problem (and solution) compared to the wireframe extraction we present. SPFN fits geometric primitives to 3D point clouds, which is sort of a "dual path" towards editable CAD models. In contrast, PC2WF abstracts a point cloud into wireframe model. I.e., we aim directly for correct topology, precise corners edges, and (for now) leave surface fitting as a "post-process"; whereas SPFN extracts parametric surfaces, and regards the (arguably more difficult) derivation of topologically correct wireframe as post-processing. Note that the output of our PC2WF is a compact, graph-structured wireframe representation whereas SPFN outputs per-point properties like point-on-primitive membership, surface normal, primitive type and primitive parameters. By construction, the number of primitive instances is fixed in SPFN, thus limiting the shape complexity that can be reconstructed. Despite these  differences, we tried to at least provide an indicative comparison here (and in the supplementary). Since the authors of SPFN did not release pre-trained models, we trained their network ourselves.  Unlike our method, training SPFN requires labels for point normals and geometric primitives, which are neither given nor easy to compute for our dataset. Instead, we trained SPFN on their own dataset (Li et al.2019), which also features mechanical components like our ABC. We then  tested the trained SPFN model on some polyhedral objects of our dataset and show qualitative results, with an accompanying discussion, in the supplementary material.

---

> ### Author Response · Authors · 2020-11-24
> **Response to Reviewer1 (w.r.t. noise level)**
>
> * In order to investigate how the noise level affects the algorithm performance, we train and test our method on point clouds with different amounts of noise. The qualitative, quantitative results and analysis are included in the updated supplementary material (Sec. D.2).

---

### Official Review · AnonReviewer4 · 2020-10-29
**Review of PC2WF: 3D Wireframe Reconstruction from Raw Point Clouds**

**Rating:** 6
**Confidence:** 5

**Review:**

### Summary
This paper introduces PC2WF, a neural network that turns 3D point clouds into a wireframe model. PC2WF encodes each point into a feature vector and uses them to predict the candidate corners. After that, line proposals are generated by connecting pairs of corners, and the point features along each line are pooled into its confidence value. By pruning the proposed lines, PC2WF generates the final wireframe represesntation.

### Comments

Extracting lines and wireframes from point clouds is a relatively new idea. However, there are lines of works about detecting 3D wireframe from single images for both indoor and outdoor scenes, in which people first detect 2D wireframes and lift them into 3d with optimization. I suggest that the author should also cite the following papers:

Zou, C., Colburn, A., Shan, Q., & Hoiem, D. (2018). LayoutNet: Reconstructing the 3d room layout from a single rgb image. In Proceedings of the IEEE Conference on Computer Vision and Pattern Recognition (pp. 2051-2059).

Zhou, Y., Qi, H., Zhai, Y., Sun, Q., Chen, Z., Wei, L. Y., & Ma, Y. (2019). Learning to reconstruct 3D Manhattan wireframes from a single image. In Proceedings of the IEEE International Conference on Computer Vision (pp. 7698-7707).

#### Pros

1. The writing of the paper is clear and the method is easy to understand. The figures are nice.

2. The end-to-end framework is intuitive and reasonable for the tasks of wireframe extraction from point clouds.

#### Suggestings/Questions

1.  This paper only tested on the synthetic ABC datasets, in which models have fairly dense point clouds and relatively low noise. It would more convencing if the authors can show whether the algorithm works on real-world 3D scanning, e.g., redwood 3d-scan.

2. It seems that EC-NET is the only data-driven method among all the baselines. Could you clarity in the experiment setting section that whether you re-train their models on your dataset with reasonable efforts? If so, could you comment on why their performance is bad (AP^e_0.01) in the experiment section?

3. Will the source code be released?

---

> ### Author Response · Authors · 2020-11-19
> **Response to Reviewer4**
>
> We thank the reviewer for the valuable suggestions. We answer all comments in detail below and add additional text, experiments, and illustrations to the main paper as well as to the supplementary material. All changes are highlighted in blue.
>
> * Thanks for the pointers, we include all mentioned papers on 3D wireframe generation from single images in the related work of the revised manuscript.
>
> * Although we did not test on real-world data, we expect our method to work well if the corresponding wireframe annotation can be obtained. The main reason for not using real 3D data is the lack of annotations, which are required to train our model. We acknowledge this is a limitation, and in future work we aim to collect usable real world data, which is very laborious and will need substantial resources. In our view it is a justifiable approach, to initially tackle a challenging topic with synthetic data; and a step forward to make progress in that setting, especially in an area with so little existing work.
>
> * We did not re-train EC-Net for the submitted version, but used the pre-trained model released by the authors. The reason for the low AP^e0.01 score may be that their model has a higher distance tolerance for edge points. When the evaluation criterion is strict, i.e., the threshold eta\_e is small, AP will penalise false positives under this strict threshold.
> When relaxing the threshold, the performance of EC-Net improves (see Tab.2). For the revised version of the paper, we re-trained EC-Net on our dataset. As one item of their joint loss function is "surface loss", we additionally use the ground truth surfaces to train EC-Net (whereas our method only needs the ground truth wireframes to train the edge point detector). The re-trained version  yields 0.619 (AP^e\_0.01), 0.860 (AP^e\_0.02), 0.922 (AP^e\_0.03), see Tab.2 in the paper. Even so, PC2WF still significantly outperforms the improved EC-Net.
>
> * We will release all code and the dataset after acceptance of the paper.

---

### Official Review · AnonReviewer3 · 2020-10-29
**This paper presents an interesting idea to form a wireframe graph for unordered pointcloud using an end-to-end network. But it doesn't have enough applications due to limitations like big training data requirement, and perfect point cloud input.**

**Rating:** 6
**Confidence:** 3

**Review:**

This paper introduced an end-to-end trainable network to predict 3D wireframes given an object point cloud using deep networks. It firstly use  network to cluster points and predict the vertex, and then use an edgenet to predict if an edge exists between two vertices.

Pros:
+ This paper presents an interesting idea on transforming an unordered pointcloud to a structured graph and use networks to learn topologies.
+ This paper presents some tricks, for example, selecting limited numbers of examples to prevent redundant computation and potential imbalance to train edgenet, and use geodesic distance for grouping points during patch formulation.
+ This paper raised a new problem and created their own synthesized dataset based on existing datasets for evaluation.

Cons:
My biggest concern is the limited application of this method. There are already many previous researches on generating graph or wireframes given point cloud without training data pairs. There are also many recent efforts on generating meshes given point cloud using deep networks. Deep networks requires tons of training data, and also seems most of the experimental results in this paper are almost perfect point cloud. Due to these limitations, I don't think this paper has enough contribution for applications.

---

> ### Author Response · Authors · 2020-11-19
> **Response to Reviewer3**
>
> We thank reviewer3 for the comments.
>
> * We did compare with one very recent method that does not follow a learning-based strategy. That method extracts planar surfaces, which can then be converted to a wireframe model. Our results indicate that PC2WF achieves significantly better results on our test set. We acknowledge that our method requires large amounts of training data, a property which it shares with other feature/representation learning methods. While in some contexts the need for training data may constitute a relevant limitation, we do not see it as an argument to rule out a learning approach, as long as the latter improves performance. Also note that in our case the large majority of training data can be generated synthetically, and thus automatically. Finally, we do contaminate our point clouds with noise, which is a standard practice point cloud processing papers, see for instance PointNet or EC-Net (Yu et al., 2018).

---

### Decision · Program_Chairs · 2021-01-07
**Final Decision**

**Decision:**

Accept (Poster)

**Comment:**

Reviewers were all on the positive side for this paper. Multiple reviewers liked the new and interesting task that this paper presents, found that the proposed method works well, is sufficiently compared to alternative approaches, and could serve as a solid baseline for future work in this area. The main limitation that reviewers noted was that results were shown only on the synthetic ABC dataset using dense point clouds with very little noise. The authors wrote *very* thorough responses to all reviewer questions. One reviewer noted that these responses answered all of their questions.

It is worth noting that a paper that solves a similar problem was recently published at NeurIPS ("PIE-NET: Parametric Inference of Point Cloud Edges"). This paper was not published as of the ICLR submission deadline, so it was judged as 'contemporary work' which the authors have no obligation to compare against. Nevertheless, they did attempt to make a comparison in their responses. I would ask that the authors include some discussion of this comparison in their final version of the paper.